# Beyond the Second Law: Darwinian Evolution as a Tendency for Entropy Production to Increase

**DOI:** 10.3390/e27080850

**Published:** 2025-08-11

**Authors:** Charles H. Lineweaver

**Affiliations:** Research School of Astronomy and Astrophysics, Australian National University, Canberra, ACT 0200, Australia; charley.lineweaver@anu.edu.au

**Keywords:** entropy, entropy production, Darwinian evolution, complexity

## Abstract

There is much confusion about the apparent opposition between Darwinian evolution and the second law of thermodynamics. Both entropy and entropy production play more fundamental roles in the origin of life and Darwinian evolution than is generally recognized. I argue that Darwinian evolution can be understood as a tendency for entropy production to increase. Since the second law is about the increase in entropy, this hypothesis goes beyond the second law because it is about the increase in entropy production. This hypothesis can explain some aspects of biology that Darwinism struggles with, such as the origin of life, the origin of Darwinism, ecological successions, and an apparent general trend towards biological complexity. Gould proposed a wall of minimal complexity to explain this apparent increase in biological complexity. I argue that the apparent increase in biological complexity can be understood as a tendency for biological entropy production to increase through a broader range of free energy transduction mechanisms. In the context of a simple universe-in-a-cup-of-coffee model, entropy production is proposed as a more quantifiable replacement for the notion of complexity. Finally, I sketch the cosmic history of entropy production, which suggests that increases and decreases of free energy availability constrain the tendency for entropy production to increase.

## 1. Biologists on Entropy

Despite Schrödinger’s efforts [1], most biologists ignore entropy because they see little or no connection between Darwinian evolution and the second law of thermodynamics. However, NASA astrobiologists are looking for life beyond the Earth, so they need a definition of life. Here is NASA’s: life is a “self-sustaining chemical system capable of Darwinian evolution” [2]. The word “self” is problematic since there is no such thing as a “self-sustaining chemical system”. Sustained chemical reactions take place when the Gibbs free energy is negative, *ΔG*(*t*) < 0. This sustained negativity can only come from sources of free energy outside the life form. Life’s openness allows free energy in, but neither a closed self nor an open self can supply a continuous source of free energy. So “self-sustaining” is an oxymoron. Also, since the Gibbs free energy *G = U − TS*, entropy (*S*) must play a fundamental role in life, as Schrödinger suggested [1].

Understanding the relationship between entropy and life involves understanding how non-life became life—how chemistry became biology. This transition does not involve the improvement or the refinement of our definitions of life. It involves their deconstruction. In other words, as we trace the evolution of life back through time and try to find its abiotic, purely chemical origins, we should expect our definitions of current life to become more and more inappropriate. So, we will necessarily have to deconstruct the current apparent boundary between non-life and life to understand the transition from chemistry to biology.


*What is important in the origin of life field is **understanding the transitions that led from chemistry to biology**. So far, I have not seen that efforts to define life have contributed at all to that understanding.*
[3]

More specifically, understanding **the role of entropy** in the transitions that led from chemistry to biology is of importance here. Entropy and entropy production play a more fundamental role in (i) the origin of life and (ii) Darwinian evolution than is generally recognized by biologists. In most authoritative books on evolution, entropy is not even mentioned. There is no entry for “entropy” in the indexes of *The Origins of Life* [4], *The Structure of Evolutionary Theory* [5], *Evolution: The first Four Billion Years* [6], or *Revolutions that Made the Earth* [7]. The definition of entropy in *What is Evolution* [8] is a crude afterthought:


**
*Entropy *
**
*The degradation of matter and energy in the universe to an ultimate state of inert uniformity. Entropy can be reached only in a closed system.*


Mayr is confusing entropy with three other concepts: (1) the production of entropy, (2) the heat death of the universe, and (3) the maximum entropy of a closed system. One of the goals of this paper is to clarify the differences between these concepts and relate them to Darwinian evolution.

Trying to understand the origin and evolution of life without understanding its fundamental dependence on the second law of thermodynamics and entropy production is futile—like trying to understand the origin and development of the United States of America without understanding the role of England. Yet, it persists. We need to do a better job convincing our colleagues of the fundamental constructive role entropy plays in the origin of life and Darwinian evolution.

Many authors still subscribe to the idea that there is some kind of opposition between entropy and evolution. Dennett’s *Darwin’s Dangerous Idea: Evolution and the Meanings of Life* [9] is an insightful description of our modern understanding of Darwinism and its importance. However, he writes:


*According to the Second Law, the universe is unwinding out of a more ordered state into the ultimately disordered state known as the heat death of the universe. What then, are living things? They are things that defy this crumbling into dust, at least for a while, by not being isolated—by taking in from their environment the wherewithal to keep life and limb together…Not just individual organisms, but the whole progress of evolution that creates them, thus, can be seen as fundamental physical phenomena running contrary to the large trend of cosmic time…It is not impossible to oppose the trend of the Second Law, but it is costly.*
[9], pp 68–69

Here is another expression of the supposed opposition between life and entropy:


*…to stay alive we need to continually eat so as to combat the inevitable, destructive forces of entropy production. Entropy kills.*
[10], p. 14–15

The relationship between the second law and Darwinian evolution is not “costly” as Dennett would have it. There is no real opposition between life and the second law any more than there is opposition between a refrigerator and the second law. Smith and Morowitz [11], p. 502, call this apparent opposition the “Darwin-Clausius non-dilemma”. Incomplete descriptions of life often invoke such a faux opposition between life and the second law. This apparent opposition stems from a long tradition of separating life from non-life, biotic from abiotic, and viewing life forms as beings independent from, and more important than, the physical environment. Thus, when it comes time to tally up the entropy budget, the focus is on tallying up the low entropy inside the life form, not on the net increase in the total entropy of the life form and the environment. Any internal entropy reduction is overcompensated by the entropy exported to the environment [1,12,13]. The apparent opposition between life and the second law comes from ignoring the fact that life forms are embedded in, depend on, and coevolved with the biosphere and Earth’s environment. If we want to understand what is going on, we need to keep track of the total entropy. By explaining this clearly, Mayr redeems himself:


*It is sometimes claimed that evolution, by producing order, is in conflict with the “law of entropy” of physics, according to which evolutionary change should produce an increase of disorder. Actually, there is no conflict, because the law of entropy is valid only for closed systems, whereas the evolution of a species of organisms takes place in an open system in which organisms can reduce entropy at the expense of the environment….*
[8]

Some authors want to finesse a compromise between life and the second law:


*…Life violates the spirit but not the letter of the second law.*
[14], p 74

One simple metaphor for the role of life in the universe is a refrigerator. A refrigerator runs on free energy. You have to plug it in. The low entropy inside a refrigerator comes at the expense of increasing the entropy outside the refrigerator. This can be tested. On a hot summer day in a sealed room, try to cool the room by opening the refrigerator door. Make sure the heat from the back of the refrigerator does not leave the room through some vent. The colder you set the refrigerator, the more watts are needed, the hotter the room gets, and the more entropy is produced. The increased consumption of free energy increases the resulting entropy [15,16].

## 2. What Is the Connection Between Entropy and Biology? Going Beyond the Second Law

Rudolf Clausius coined the word “entropy” and the second law of thermodynamics: “*The entropy of the universe tends to a maximum*” or *dS* ≥ 0 [17]. The second law is the only law of physics that has a “≥” instead of an “=”. The passage of time—even the definition of time—has been attributed to this “≥” and the irreversibility it represents. The phrase “*tends to a maximum*” begs the following questions: How strong is this tendency? How quickly does entropy tend to a maximum? Is the increase monotonic? What mechanisms control the rate of entropy increase (*dS/dt*)? Can the second law be extended to address these questions? And if so, what role does life play? To answer these questions, we need to go beyond the second law.

*The two fundamental laws of thermodynamics are, of course, insufficient to determine the course of events in a physical system. They tell us that certain things cannot happen, but they do not tell us what does happen*.[18], p 151

Many researchers have discussed the connection between entropy and biology [1,15,16,18,19,20,21,22,23,24,25,26,27,28,29,30,31,32,33]. Over the past few decades, some researchers have explored the landscape beyond the second law [34,35,36,37,38,39,40,41,42,43,44,45,46,47,48,49,50]. The question of going beyond the second law often takes the form: Is there anything fundamental or at least useful that we can say about the rate of entropy increase? In other words, can we go beyond Clausius’ second law,

**(2A).** 
*the **entropy** of the universe tends to a maximum. (i.e., there is a tendency for dS/dt ≥ 0),*


toward a new hypothesis in which “entropy” is replaced by “entropy production”:

**(2B).** 
*the **entropy production dS/dt** of the universe tends to increase (i.e., there is a tendency for d^2^S/dt^2^ ≥ 0)?*


2A and 2B are distinct, independent statements. Just as there is an important distinction between position *x*, velocity *dx/dt*, and acceleration *d*^2^*x/dt*^2^, there is an equally important distinction between entropy *S*, entropy production *dS/dt*, and changes to entropy production *d*^2^*S/dt*^2^. If position *x* is increasing, then *dx/dt* is positive. But that says nothing about the rate of change of *dx/dt*. On a bike ride to the store, you can have a positive velocity, *dx/dt* > 0, but you can go fast, then slow, then speed up and still have a positive velocity. Sometimes you push harder on the pedals and sometimes you brake. These changes in velocity are described by acceleration: *d*^2^*x/dt*^2^. Speeding up (*d*^2^*x/dt*^2^ > 0) and slowing down (*d*^2^*x/dt*^2^ < 0) are changes in the sign of *d*^2^*x/dt*^2^. Similarly, statement 2B above is about a tendency for entropy production to increase: “*a tendency for d*^2^*S/dt*^2^ ≥ 0”. It is distinct from statement 2A (the second law): “*a tendency for dS/dt* ≥ 0”. In the bike analogy, 2A is a tendency to go in one direction with no mention of speed, pedaling, or braking, while 2B is a distinct hypothesis that goes beyond the second law by suggesting there is a tendency for more pedaling and less braking. Some authors have expressed concepts similar to 2B:


*In the jargon of thermodynamics, the formation of patterns in these [complex emergent] systems helps to speed up the dissipation of energy as mandated by the second law.*
[51], p. 13

In this quote, “dissipation of energy” means *dS/dt* > 0 and is the second law (2A). But the full quote, “speed up the dissipation of energy”, is different. It goes beyond the second law and is similar to hypothesis 2B. 2B involves the second derivative *d*^2^*S/dt*^2^, not the first derivative *dS/dt*. An increase in entropy *dS/dt* > 0 is different from an increase in entropy production *d*^2^*S/dt*^2^ > 0. The “dissipation of energy” involves *dS/dt* > 0, while “speeding up the dissipation of energy” involves *d*^2^*S/dt*^2^ > 0. Thus, Hazen is assuming an idea about thermodynamics that is not “mandated by the second law”. It goes beyond the second law. The second law is completely silent on the second derivative of entropy. In Clausius’ description of the second law *(“the entropy of the universe tends to a maximum”)*, there is nothing about far-from-equilibrium dissipative systems (“FFEDS”) [52,53,54] and specifically nothing about speeding up the dissipation of energy. No part of the second law is about speeding up the rate of entropy production. If going beyond the second law involves speeding up the rate of entropy production, then something like hypothesis 2B would be a compact way to say it.

In 2A, the “*tendency for dS/dt* ≥ 0” is not a monotonic tendency. Statistical fluctuations can transiently produce *dS/dt* < 0 [55]. Similarly, in 2B, the “*tendency for d*^2^*S/dt*^2^ ≥ 0” is not a monotonic tendency. Statistical fluctuations can transiently produce *d*^2^*S/dt*^2^ < 0. In addition, the seasonal ups and downs of free energy availability produce increases and decreases in *dS/dt*. For example, weather conditions in hurricane alleys control the creation and destruction of hurricanes, increasing and decreasing the entropy production of the hurricane. Free energy from chemical and thermal gradients depends on the time of day or the season. These FFEDS-driving gradients are sometimes interrupted by large asteroid impacts or drastic changes in climate. Thus, the entropy production of FFEDS tracks the varying availability of free energy without any obvious tendency for a secular increase in entropy production.

There are other problems involved with efforts to go beyond the second law:

(A) Physicists often insist that the entropy of a system can only be defined in equilibrium. If the concept of entropy only makes sense in equilibrium, we can only talk about the difference in the entropy between two equilibrium states, *ΔS = S*_2_ − *S*_1_, or about near-equilibrium states [11,37,56,57,58,59,60].

(B) A diversity of mathematical notations and an acronym soup (e.g., MEP, MaxEP, MinEP, MaxEnt) have evolved from different approaches and attempts to generalize our understanding of entropy production to the widest range of applications. For example, Table 1, Figure 1.1, of [49] uses the dissipation function <Ω> to mathematically explore the landscape beyond the second law. Bruers [40,41,42] discusses various versions of MEP depending on whether the entropy production is near equilibrium or far from equilibrium. His “far-from-equilibrium variational MaxEP” seems to be the most relevant to keeping track of biological entropy production. Other researchers describe entropy production in the context of biochemical reactions [61], chemistry [50], molecular biology [62], planetary atmospheres [63], and the Gaia hypothesis [64,65,66,67]. The validity and applicability of these various entropy production principles seem to depend on the problem and on the mechanism of entropy production. Thus, currently, there is no general-purpose principle for systems far from thermodynamic equilibrium.

(C) In analyses of cosmological entropy, the entire observable universe (or a large representative volume) is the system under consideration. We denote its entropy by *S*. Because the system is surrounded by identical systems, there is no net import or export of entropy into or out of the system. The FFEDS discussed here are sub-systems, each of which makes a *net* contribution to the entropy production in the universe. These contributions can vary in magnitude but are always greater than or equal to zero. For details on this cosmological convention, see [68], pp. 699–700; [69], pp. 415–416; and [70].

## 3. Paradigm Shift from “We-Eat-Food” to “Food-Has-Produced-Us-to-Eat-It”

There is more to the relationship between life and entropy than life not violating the second law. Some researchers have argued that a paradigm shift is needed to more accurately describe the relationship between life and entropy—a paradigm shift from the conventional life-centered view of “we-eat-food” to a more objective gradient-centered view of “food-has-created-us-to-eat-it” [15,29,54,71].


*…gradients, when steep enough, give rise to far from equilibrium dissipative structures (e.g., galaxies, stars, black holes, hurricanes and life) which emerge spontaneously to hasten the destruction of the gradients which spawned them. This represents a paradigm shift from “we eat food” to “food has produced us to eat it.*
[54]

In the “we-eat-food” paradigm, food costs us something, and organisms that have evolved by Darwinian evolution are fighting against the increase in entropy to stay alive. In contrast, the “food-has-created-us-to-eat-it” emphasizes the intrinsic dependence of Darwinian evolution on the second law. In this view, life (as well as abiotic FFEDS) is produced and driven by chemical and thermodynamic gradients. We owe our existence to such gradients. And as we come into existence, we destroy the gradients that gave birth to us [15,31]. Abiotic FFEDS can easily be interpreted this way. Destroying gradients is what hurricanes are doing in the Caribbean, what forest fires are doing in the Pacific Northwest, what lightning is doing in Jupiter’s atmosphere, what convection cells are doing on the surface of the Sun, and what fusion reactions are doing in stellar cores.

Our existence, and the existence of far-from-equilibrium dissipative systems (FFEDS) in general (both abiotic and biotic), increases the entropy production of the universe over and above what it would be if they did not exist. This non-flattering new paradigm goes beyond the second law. In this new paradigm, life is not costly. Life forms are created by gradients to destroy those gradients and, in the process, produce more entropy than would be the case without them. To paraphrase Lovelock and Margulis [72]: Life exists “by and for” entropy production. Hoffman [73] has a slightly different idea:


*Life does not exist despite the second law of thermodynamics; instead, life has evolved to take full advantage of the second law wherever it can.*


This is an expression of the conventional “we-eat-food” paradigm and differs from food-has-created-us-to-eat-it. For Hoffman, life is a free agent with free will that can take advantage of a situation. In the “food-has-created-us-to-eat-it” paradigm, life (and FFEDS in general) are not taking advantage of a situation. Rather, they are forced into existence by a gradient to reduce that gradient and increase entropy. To underline how fundamental a change in perspective this paradigm shift is, consider how Dyson, in his well-cited article “Energy in the Universe” [74], described the role of life in the universe:


*…life may have a larger role to play than we have yet imagined. Life may succeed against all odds in molding the universe to its own purposes.*
[74], p. 51

In contrast to Dyson, I am arguing that at the most fundamental level, life does not mold “*the universe to its own purposes*”. Rather, life has been molded by physical and chemical gradients to destroy those gradients and produce entropy. If life has a “*role*” or “*its own purpose*”, it is the production of entropy. And rather than this being “*against all odds*”, it is a definitional feature of entropy increase that the odds are in its favor.

## 4. Enhancement Compared to What?

FFEDS are conventionally divided into two classes: (1) abiotic and (2) biotic. Examples of abiotic FFEDS include convection cells, hurricanes, whirlpools, lightning, stars, and galactic bars. These are caused by gradients of various kinds: thermal, pressure, humidity, photochemical, electric, chemical redox, and gravitational [54,74]. The earliest life forms were chemoautotrophs produced by chemical gradients [75,76,77,78]. These chemical gradients were produced by gravitationally induced density and thermal gradients. Life is still powered by chemical gradients, but predominantly by photon-energy gradients. A hierarchy of transduction mechanisms transfers power from one gradient to another [44,45,54,79]. When confined to biotic FFEDS, this hierarchy is called a trophic pyramid.

Both abiotic and biotic FFEDS are born from gradients. Both produce entropy. Despite the entropy production of abiotic FFEDS, it is problematic to say that their “existence increases the entropy of the universe over and above what it would be if they did not exist”, because we have no idea what a universe without convection, turbulence, hurricanes, fires, or stars would be like. Thus, it makes little sense to use such a contrafactual universe to calibrate the entropy production of abiotic FFEDS. In contrast, we can more easily imagine a universe without life—without biotic FFEDS. In which case, it seems reasonable to state that biotic FFEDS (e.g., microbes, mushrooms, humans) increase the entropy of the universe over and above what it would be in their absence. Thus, compared to an abiotic universe, a biotic universe will have larger entropy production since such a universe contains (but not out of necessity) evolving, free-energy-tapping, entropy-producing organisms who, despite their low internal entropy, contribute a net positive tally to the total entropy of the universe. This is illustrated in Figures 3 and 4 by the green lines (with life) being above the black lines (without life).

## 5. The Origin of Life: The Emergence and Evolution of Biotic FFEDS from Abiotic FFEDS

Abiotic FFEDS emerge or develop (some even say “evolve” [80,81]) from gradients. Some emerge rapidly, like the branching patterns of a lightning bolt; some more slowly, like the turbulent whirls as cream mixes with coffee. Some emerge and develop on longer time scales of days or thousands of years (hurricanes, river basins, Jupiter’s great red spot). As abiotic FFEDS come into and go out of existence, they do not accumulate information. They have no coded DNA. They have inertia, but no controllable stored energy. If they “evolve”, their evolution is not a result of Darwinian evolution. For example, the relationships between hurricane formation and gradients of temperature, density, pressure, and humidity are the same now as they were 100 million or a billion years ago and are the same in our galaxy as they are in other galaxies. Ditto for dust devils, convection cells on stellar surfaces, and the fusion of H into He inside stars.

Biotic FFEDS must have emerged from abiotic FFEDS by stumbling upon energy transduction mechanisms [11,78,79,82] that enabled a rudimentary form of energy storage. This gave biotic FFEDS the ability to persist through periods of free energy shortages. Many free energy gradients are sporadic and temporary. The Sun shines only during the day. When the gradients disappear, so must the abiotic FFEDS. Biotic FFEDS, because of their ability to store energy, are more robust and less easily extinguished by a non-continuous supply of free energy. Also, temporary or seasonal variability of free energy availability led to the evolution of periods of inactivity, bacterial dormancy [83], sporulation, and hibernation. Hurricanes cannot hibernate. To help distinguish between abiotic and biotic FFEDS, we adopt a definition of life suggested by Smith 2008 [84]: Life is a chemical system in which the flow and storage of energy are related to the flow and storage of information. The flow and storage of energy and information are continuous variables that can start at arbitrarily low levels. Thus, this definition can plausibly describe a continuous transition from non-life to life [85,86]. In this scenario, the earliest form of replication would be the spontaneous prebiotic autocatalysis that leads to the evolution of catalysts and the enhancement of kinetic control [86,87].

Biological evolution discovers ways to access new sources of free energy that abiotic FFEDS cannot access. Compared to abiotic reaction rates, organic catalysts increase the rates of chemical reactions by many orders of magnitude—allowing and controlling reaction rates that would never occur abiotically. The Darwinian evolution of these catalysts increases the rate even more. Through these accumulated adaptations, biological evolution comes up with new sources of free energy and new ways to make free energy more available and therefore new ways of increasing *dS/dt* [31]. Life forms (“biotic FFEDS”) can be understood as catalysts that, as they evolve, lower the activation energies for their metabolic reactions [88,89]. Evolving catalysts allow flexible, increased availability of free energy and increase entropy production [90].

On a global level, as biogeochemical cycles persist, they increase their entropy production by evolving a more comprehensive hierarchy of mutually dependent, free-energy-transducing, entropy-producing mechanisms [33,61,65,91,92,93,94,95]. The emergence of life opened new channels for chemical energy flow on Earth [11]. The ~3-billion-year evolution of photosynthesis is one example of the opening of such new channels [96].

## 6. A Tendency Towards Increased Entropy Production Can Explain the Apparent Increase in Biological Complexity

Although complexity is difficult to define and a multitude of definitions exist [97,98], many biologists and most people think Darwinian evolution has an inbuilt direction towards increasing complexity:


*Stars are the power plants that drive life’s development toward increasing complexity.*
[99], p. 159

To many, the human brain is the most complicated thing in the universe, followed close behind by our computers, cities, and social networks. “*Once there were bacteria, now there is New York*” [100]. Most people are convinced that, contrary to what one would expect from the second law, the complexity of the universe is increasing. This is part of the apparent opposition between life and entropy discussed in Section 1. This conviction of an increase in complexity is usually associated with the evolution of multicellular life forms and the other “major transitions” in the evolution of life [4]. Darwin, however, seems to have been non-committal about whether evolution had a direction towards increased complexity [101].

One problem with the view that complexity is increasing is that the complexity of some species increases while the complexity of other species decreases. For example, for every complex multicellular organism, a dozen species have evolved to become simpler and parasitic and live off the multicellular organism with its convenient flow of free energy. If we are counting the difference between the number of life forms that have become simpler and the number that have become more complex, the number getting simpler is easily larger. Ignoring parasites and paying more attention to the evolution of the most complex organisms gives the impression that complexity is increasing. The increase in biological complexity and the simultaneous increase in biological simplicity are not a contradiction. Both can be explained by the unifying concept of biological evolution as the product of a tendency to increase *dS/dt* (2B). The apparent increase in biological complexity can be better understood as a tendency for biological entropy production to increase, resulting in a broader range of both large and small mechanisms of free energy transduction.

## 7. Gould’s Wall of Minimal Complexity

The idea that the complexity of life is increasing is a conventional—even dominant view—of how Darwinian evolution works. In an effort to push back on this idea, Gould [102] introduced the idea of a wall of minimum complexity (the red brick wall in panels B and C of Figure 1).


*For reasons related to the chemistry of life’s origin and the physics of self-organization, the first living things arose at the lower limit of life’s conceivable preservable complexity. Call this lower limit the ‘left wall’ for an architecture of complexity. Since so little space exists between the left wall and life’s initial bacterial mode in the fossil record, only one direction for future increment exists—toward greater complexity at the right.*
[102]

Gould’s view was that such a one-sided wall left nothing else for a diversity-prone evolutionary process to do but extend to the right. This wall of minimal complexity was effectively a wall of irreversibility in the sense that early life could evolve through it, going from left to right, but evolved life could not go back (right to left). This wall is a visual expression of Dollo’s law of irreversibility: an organism never returns exactly to a former state [103,104].

There are several problems with this wall of minimum complexity. One is that since there are many ways of being complex [97], complexity should not be represented as one-dimensional (as it is in Figure 1). One-dimensional representations of “intelligence” are similarly misleading [105]. Another problem is that since we do not know how life emerged 4 billion years ago in the Hadean, we do not know whether the emergence was irreversible. Abiotic FFEDS seem to be able to form and disappear—to become more complex and less complex with no concern for any wall of minimal irreversible complexity preventing them from disappearing. Any one-way wall at some minimal complexity required for life would depend on life not being able to reduce the amount of accumulated information. However, four billion years ago, when the amount of accumulated information in the earliest biotic FFEDS was small, the degree of irreversibility would also have been small. Since Dollo’s law [103,104] must depend on the amount of accumulated information, the strength of any irreversibility would have been diminished in the Hadean.

Another problem. To justify the wall, Gould invoked the idea that “*so little space exists between the left wall and life’s initial bacterial mode*”. In Gould’s world of Cambrian and post-Cambrian paleontology, there may be “little space” between the wall and bacteria, but in the past few decades, microbiology has filled this “little space” with a huge variety of organisms at the boundaries between life and non-life. These include a major new branch of largely symbiotic and episymbiotic bacteria [106,107,108] as well as new types of viruses: mimiviruses and megaphages [109,110,111]. Half of biologists think viruses are alive while the other half think they are not. Whatever they are, they are the most abundant organisms on Earth and were possibly fundamentally involved with the origin of life [112] (https://www.youtube.com/watch?v=Orf_cbXyZHo&t=468s, accessed on 10 April 2025). Do we put viruses to the left or right of the wall of minimal complexity? And what about free-living viroids [113]? Are McClintock’s transposons (=jumping genes) alive or not alive? “*Every component of the organism is as much of an organism as every other part*” [114]. Along with the Hadean RNA world [115,116], there is a large unexplored and unappreciated extant RNA world [117,118,119] that seems to straddle any supposed wall of minimal complexity.

In describing his wall, Gould writes: “*…little space exists between the left wall and life’s initial bacterial mode in **the fossil record***”. But the Precambrian microbial fossil record is so sparse as to be irrelevant. Bacteria do not preserve well. We have almost no rock fossils of the Last Universal Common Ancestor (LUCA) and do not expect to find any. Our ideas about the time span between the origins of life and LUCA are speculative. However, most of the literature on the topic admits that LUCA was already so complicated that a substantial amount of time would have been required for the evolution of such complexity [77,120]. We also need to consider pre-cellular life. Do organisms need cell walls, DNA, or RNA to be on the right side of the wall? Do metabolism-first models live only on the left [121,122]? In the absence of a wall of minimal complexity, entropy production explanations for the simultaneous increase in biological complexity and biological simplicity seem more plausible than Gould’s one-way wall. There are, however, alternative views on the apparent increase in complexity and entropy production [123,124,125].

## 8. Techno FFEDS

We humans learned to control fire. This gave us the ability to cook and access more free energy in a wider variety of foods [126]. This new free energy probably freed the energy-intensive brain from some constraints. Our brains became bigger and figured out how to burn hydrocarbons, make solar panels, wind mills, and atomic fission reactors [127]. Soon fusion reactors may tap into the free energy of hydrogen just as abiotic FFEDS (stars) began doing 13.6 billion years ago [128]. Eventually, we may extract free energy from rotating black holes via the Penrose mechanism [129]. In a speculative continuation of this pattern into the future, new sources of free energy may release energy-intensive silicon-based artificial intelligence (‘techno FFEDS’) from some energy constraints. Kardashev has proposed a way to classify future civilizations [130,131]. Kardashev Type I civilizations access all the free energy available on a planet. Type II access all the free energy available from its host star. Type III access all the free energy available from the hundreds of billions of stars in a galaxy. Even more speculatively, much larger supplies of free energy may enable the construction of anti-matter-powered rockets [132], Dyson spheres [133], and possibly whole universes [134]. All these may give life forms access to more free energy, thus increasing the entropy production attributable to biological (and post-biological) evolution. This access to free energy has the potential downside of increasing entropy production too fast. Civilizations could suffer from unsustainable growth [16].

## 9. Entropy Production as a Useful Expression of ‘Complexity’

In Figure 2 and the discussion of the mixing of coffee and cream, we are referring to the dynamical complexity of the intricate multi-scale turbulent eddies, e.g., [135]. This is comparable to the morphological complexity of salt fingers and to other patterns formed by other processes in non-equilibrium systems such as hurricanes, thermal convection, shear turbulence, and oceanic circulation, in which the rate of energy dissipation and entropy production increase as the structure develops [136].

Since complexity is notoriously hard to define [97,98,139,140,141], entropy production may be a potentially useful, more quantitative measure that still contains some of the intuitive notions associated with ‘complexity’. This replacement is illustrated in the bottom panel in Figure 2 and is quite different from the middle panel [137,138]. One test of whether entropy production is a reasonable replacement for complexity is to compare the two in situations where complexity is well-defined. For example, one could take a video of the mixing process in the top panel of Figure 2 and, using a file-compression algorithm, quantify the Kolmogorov algorithmic complexity of the turbulent swirls [138], p. 232, [142]. The time dependence of the Kolmogorov algorithmic complexity in the video can then be compared to the time dependence of the entropy production of mixing.

We assume that the universe started out at very low entropy. This is the “Past Hypothesis” [143,144]. We know almost nothing about the universe before inflation. Inflation expanded and cooled whatever was there. The first and steepest increase in the entropy of the universe (when *dS/dt* was at its maximum value) happened at the end of inflation during reheating ~10^−35^ s after the Big Bang. During reheating, the potential energy of the inflaton field decayed and became coupled to, and excited, all the other quantum fields. The excitation of these fields was the creation of all the matter and energy in a hot, dense universe. Thus, the one degree of freedom of the potential energy of the inflation scalar field became distributed over all the particles in the universe. This spreading out of the potential energy to many more degrees of freedom is what increased the entropy of the universe. After reheating, the second steepest increase in entropy (the second largest value of *dS/dt*) occurred between 11 billion years ago and 9 billion years ago. Quasar observations suggest that this is the time of peak supermassive blackhole (SMBH) formation [145]. SMBHs are the largest reservoir of entropy in the universe [146]. They did not exist at the Big Bang or at recombination, so they acquired their large entropies more than a billion years after the Big Bang.

Comparing the black and green lines in Figure 3 illustrates the idea that life forms (biotic FFEDS) increase the entropy of the universe more than would be the case if they did not exist. As suggested in the top right of Figure 3, the entropy produced by life initially makes a tiny contribution to the total entropy. See [70,146] for details of the abiotic history of entropy in the universe.

In Figure 4, we see *dS/dt* going up and down, increasing and decreasing. Thus, hypothesis 2B does not describe entropy production on cosmic time scales. Although the entropy production of abiotic FFEDS is strictly constrained by what can be called the “abiotic availability of free energy”, the entropy production of biotic FFEDS is able to partially exceed these abiotic constraints due to the ability of life to access, store, and use new sources of free energy, thus increasing entropy production above what it would be in the absence of life (green line in Figure 4). While biotic FFEDS exceed abiotic limits, what is or is not biotically available free energy depends on evolutionary paths, historical contingencies, and the open-ended nature of Darwinian evolution [5]. Thus, the limits of biotically available free energy are not well-defined and cannot be easily predicted. This is an obstacle to articulating a general-purpose entropy production principle.

There is some agreement that life forms enhance the operation of the second law [15], p. xv. That is, biotic FFEDS make *dS/dt* larger than it would be without them. With this in mind, we should reduce the generality of hypothesis 2B by restricting it to biotic FFEDS. In other words, we should change 2B from:

**(2B).** 
*the entropy production dS/dt of the universe tends to increase (i.e., there is a tendency for d^2^S/dt^2^ ≥ 0)*


to

**(2B’).** 
*the entropy production of life forms dS/dt_bio_ tends to increase (i.e., there is a tendency for d^2^S/dt_bio_^2^ ≥ 0).*


Qualitatively, this makes sense. However, trying to quantify this enhancement is a work in progress [90,147]. Combining this tendency for *dS/dt_bio_* to increase with the proposal that *dS/dt (t)* is a useful expression for ‘complexity’ yields the plausible conjecture that a tendency for biological entropy production to increase is consistent with, but is a more quantifiable concept than, an apparent tendency for biological complexity to increase.

Although we have emphasized the second law of thermodynamics, the biological entropy production of 2B’ involves kinetics, not just thermodynamics. Activation energies and reaction rates are kinetically controlled by catalysts, not by thermodynamics and the ΔG to the ground state [148] (Chapter 8). Over billions of years, enzymes have evolved to not only accelerate reactions but also to organize and control them so that much of the energy released is recovered in other chemical forms, which eventually results in increased entropy production [148,149,150].

## 10. A Tendency to Increase Entropy Production Can Resolve Some Problems with Darwinian Explanations

A tendency for entropy to increase can explain some aspects of biology that Darwinism struggles with. Arguably, the most important examples are Darwinism’s inability to explain the origin of life, the origin of Darwinism [151], or the apparent general trend towards biological complexity [11,97]. Evolution by natural selection needs a target or a “unit of selection”. In post-Cambrian evolutionary biology, the focus has been on the multicellular individual as the unit of selection [5,152,153]. However, in many situations, the unit of selection is not well-defined. Is the unit of selection the gene [154], the group [153], the clade [152,155,156], or some multilevel combination [157]? Explanations based on an increase in entropy production are available to explain the origin of life and the origin of Darwinism [11,31,79,82]. Such entropy-production explanations are not compromised by “unit of selection” issues.

Darwinism also struggles to explain large-scale biological patterns such as ecological successions, the pole-to-equator increase in the diversity of species, and the increasing species diversity and complexity of ecological networks that make them more stable. On the largest biospheric level, explaining Gaian regulation as the result of evolution by natural selection is also problematic [95,158]. Selection based on a tendency for increased entropy production works at all levels of biological organization.

Another potential problem of Darwinian explanations is the extreme wastefulness of “trial and error”. This waste is usually rationalized as a way to increase selection pressure, thereby making the few survivors more fit. Do we have evidence that *all* such death and wastefulness have evolved to improve fitness? Could the Malthusian fitness-based explanation (i.e., we need this much death to keep selection pressure high) be only partially correct [159]? A tendency to increase entropy production is obviously consistent with such waste. All biological activity, whether wasteful or not, produces entropy. If the activity increases on evolutionary timescales, this increases entropy production. Notice that 2B’ does not explicitly distinguish between the “use” or “waste” of the newly accessible free energy that life forms have evolved to access. However, if it is “used” (i.e., if the free-energy-transducing mechanisms can channel some of the free energy into accessing more free energy), this would increase entropy production. Thus, “using” (rather than wasting) the free energy provides a feedback that could support the tendency for biological entropy production to increase.

The continuing survival of a life form is necessary to allow it to continue to produce entropy. Thus, “survival of the fittest” and “the increase of entropy production” can both be considered as complementary functional objectives in biological evolution [61]. For example, “survival of the fittest” could be the proximal explanation, while “the increase of entropy production” could be the ultimate explanation.

For several decades, “self-organization” has been proposed as an alternative to Darwinian explanations for the origin of biological order [5,53,160,161]. Do self-organized structures emerge at specific values of the entropy production rate [162,163]? Self-organization is “an expression of properties intrinsic to complex dynamic systems organized by simple rules of interaction among large numbers of elements” [161], p. 186. Hypothesis 2B’ suggests that these intrinsic properties and “simple rules of interaction” may be usefully described by a tendency towards the increase in biological entropy production.

Long-term experiments in microbial evolution have shown that microbes evolve the capacity to use previously untapped resources [164,165]. It is possible that such experiments could be modified to test the claim that the tendency for *dS/dt_bio_* to increase is independent of Darwinian “survival of the fittest”. For example, in a chemostat where chemical and thermal entropy can be quantified, supply bacteria that can only metabolize glucose with nutrients containing glucose, sucrose, and lactose. Selection pressure can be minimized or eliminated by keeping the glucose-based carrying capacity of the nutrient substantially above the glucose needs of the population [166]. The prediction of the “tendency for *dS/dt_bio_* to increase” model is that the bacteria will diversify and develop the ability to access the free energy in the sucrose and lactose (in addition to the glucose), thus increasing entropy production in the absence of selection pressure.

## 11. Summary

By emphasizing the difference between entropy and entropy production, the meaning of “going beyond the 2nd law” is clarified. This helps resolve some of the widespread confusion about the relationship between evolution and entropy. The concept of far-from-equilibrium dissipative structures (FFEDS) is used to clarify the relationship between entropy production and life. It suggests a paradigm shift from “we eat food” to “food has created us to eat it”. As a way to describe the role of FFEDS in going beyond the second law, hypothesis 2B was proposed: *the entropy production dS/dt of the universe tends to increase*. However, it was found wanting because the entropy production of abiotic FFEDS is constrained by the ups and downs of abiotically accessible free energy. Biotic FFEDS are able to access, store, and use more than just abiotically accessible free energy. Based on this ability, a modified hypothesis 2B’ was proposed: *the entropy production of life forms dS/dt_bio_ tends to increase.* It incorporates the idea that life (biotic FFEDS) is sustained biochemical entropy production, and biological evolution is a tendency for that production to increase. Gould’s wall of minimal complexity is criticized based on the entropy production argument for the origin of life and the wide range of newly discovered organisms that would straddle this wall (if it existed). In the context of a simple cream-in-coffee model of cosmic entropy production, I propose that entropy production can replace the concept of ‘complexity’ while still maintaining many of our intuitive notions about complexity. I argue that the apparent increase in biological complexity is subsumed within the broader idea that there is a tendency for biological entropy production to increase.

## Figures and Tables

**Figure 1 entropy-27-00850-f001:**
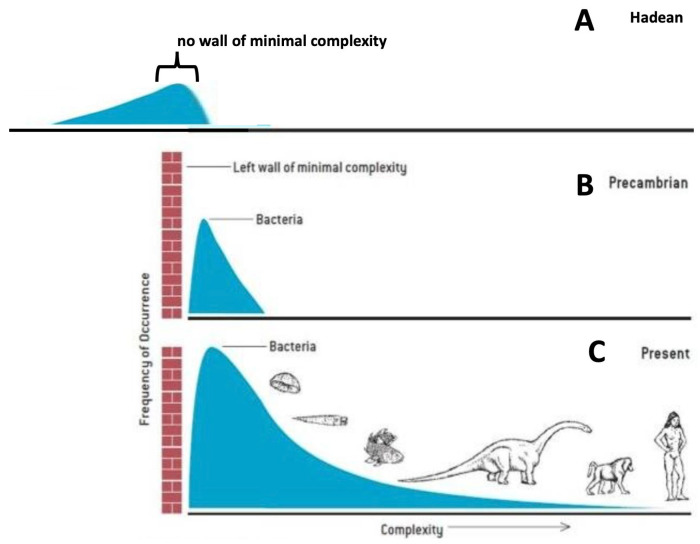
Questioning Gould’s wall of minimal complexity. Gould introduced the idea of a wall of minimum complexity (panels (**B**,**C**)) to explain that, although Darwinian evolution diverged towards both more complexity and more simplicity, it seemed to produce only increasing complexity [102]. In Panel (**A**), the Hadean period, about 4 billion years ago, represents the idea that there was no minimum wall of complexity. In the text, I argue that there never was, and there currently is not, such a wall.

**Figure 2 entropy-27-00850-f002:**
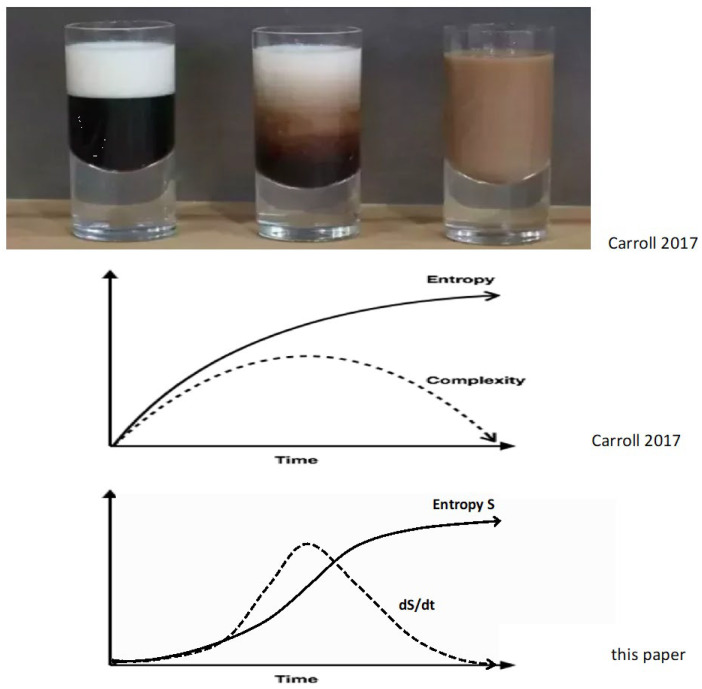
Coffee, cream, entropy, and complexity. The top panel shows three stages in the process of cream mixing with coffee [137,138] (Chapter 28). In the left image, the cream and coffee are separate and unmixed: low entropy. In the middle image of the top panel, complex convection, turbulent swirls, and diffusion mix the cream and coffee. In the right image, the cream and coffee are completely mixed at equilibrium and maximum entropy. Although complexity is famously hard to define, just below the three photos is Carroll’s notional plot of the relationship between entropy and complexity during this cream/coffee mixing process. The entropy starts low and ends high. On the other hand, the complexity of the convective swirls starts low, reaches a maximum when the most complicated range of swirls is most effectively mixing the cream with the coffee, and then complexity decreases as the swirls disappear. The lowest plot is an alternative view of the relationship between entropy and complexity. It assumes that entropy production *dS/dt* can subsume or replace some of our notional views of ‘complexity’. This plot is based on the idea that the more complex the convective swirls and fluid turbulence are, the more entropy is being produced—both by the undoing of the concentration gradients by the swirls and by the free energy needed to maintain the swirls. By way of contrast, in Carroll’s notional plot, the rate of increase in ‘complexity’ is implausibly a maximum at the very beginning of the mixing when most of the convection and swirling eddies have not yet developed.

**Figure 3 entropy-27-00850-f003:**
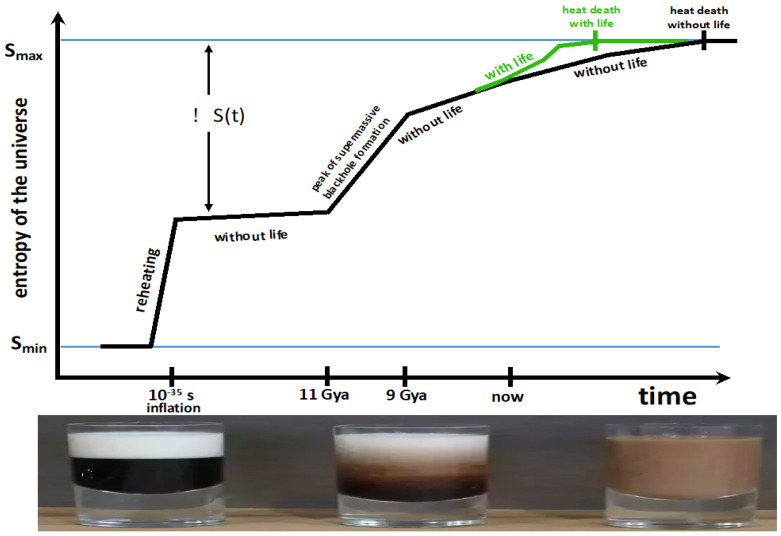
The universe in a cup of coffee. Top panel: The entropy of the universe as a function of time without life (black line) and with life (green line). The universe starts out on the left at low entropy *S_min_* and ends up on the right at an equilibrium heat death of maximum entropy *S_max_*. Δ*S*(*t*) is a measure of the number of degrees of freedom which will eventually, but have not yet, become accessible. I have assumed that the total number of eventually accessible degrees of freedom will not depend on whether life exists or not. As life evolves, it tends to produce more effective catalysts, making more free energy available, thus increasing the entropy of the universe compared to a universe without life. Thus, a universe with life (green line) reaches a heat death (*S_max_*) sooner than a universe without life. As the universe approaches the heat death, there are fewer gradients that can maintain life forms or the abiotic sources of entropy production. Thus, for both the green and black lines, the slope *dS/dt* → 0 as the universe approaches its heat death. Bottom panel: This cream/coffee sequence from Figure 2 is a model for the past, present, and future entropy production in the universe. The universe starts out at low entropy (unmixed cream and coffee). Then, concentration, thermal, and kinetic gradients between the cream and coffee produce a complex mixture of convective swirls that mix the cream and coffee, thus undoing the gradients. When these gradients are gone, we have an equilibrium or heat death in which there are no gradients to support FFEDS. A gaggle of the most advanced Kardeshev Type III civilizations might be responsible for the steepest slope (=largest value of *dS/dt*) of the green line. This sketch is based on Figure 22.2 of [69] and Figure 6 of [146].

**Figure 4 entropy-27-00850-f004:**
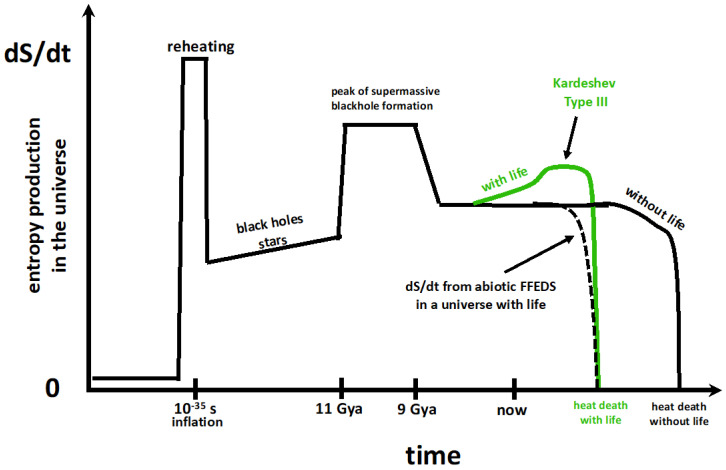
Time dependence of cosmic entropy production. The ups and downs in this plot are associated with the increase then decrease in *dS/dt* during reheating and with the increase then decrease in *dS/dt* caused by the increase then decrease in the formation rate of stars and blackholes. Since *dS/dt* ≥ 0 everywhere in this plot, it conforms with the second law (2A). Since there is no obvious tendency for *dS/dt* to increase over cosmic history, it does not conform to hypothesis 2B. However, the effect of life (represented by the green line) does conform to 2B until the approaching heat death of the universe (hastened by the existence of life) begins to turn off all supplies of free energy for abiotic FFEDS, biotic FFEDS, and any advanced civilizations (techno FFEDS). This approach to the heat death puts a final upper limit (peak labeled “Kardeshev Type III”) to the increasing entropy production due to the presence of life. For details, see Figures 6 and 7 of [146].

## Data Availability

Data are contained within the article.

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
