# Peer review of "Beyond the Second Law: Darwinian Evolution as a Tendency for Entropy Production to Increase"

_entropy, 2025, doi:10.3390/e27080850_

Round 1
Reviewer 1 Report (New Reviewer)
Comments and Suggestions for Authors
The preset manuscript is not clear in the present form what the author tried to clarify for the relation between Darwinian evolution and Entropy production in the present manuscript.
At the end of summary, in line 630-632, the author proposed "by a simple cream in coffee model an operational definition in which complexity is proportional to dS/dt".
Comments of the reviewer are mostly summarized by focusing on this sentence, although related descriptions are found all over the manuscript.
First of all, what part of entropy change the author means by dS/dt ? Far from equilibrium dissipative system (FFEDS) is generally understood as a local steady system in which a dynamical structure constantly creates entropy, and simultaneously keeps a low entropy state by constantly throwing out this produced entropy out into the reservoir. This is only possible when the local system is immersed in a low entropy reservoir (This is why it is called Far From Equilibrium.) which is open to accept the produced entropy for carrying out into the atmosphere. Low entropy local system with a stable dynamical structure which keeps producing entropy is the essential character of FFEDS, whether it is biotic or abiotic.
The "cream in coffee model" is a closed system and difficult to be used for explaining FFEDS.
Secondly, "complexity" is not defined in the manuscript. How can an undefined quantity is connected to dS/dt by a mathematical word "proportional"?
Therefore, this sentense in the summary does not seem to be meaningful.
Reviewer believes that the present research will become fruitful in future, when the author clarifies how he thermodynamically defines entropy, whether he defines it for each biological body or for some groups or total biomass.
Also, it would be necessary for the author to define complexity after clarifying whether he wishes to study complexity of a biological body or for the ensemble of biologies.
Author Response
The preset manuscript is not clear in the present form what the author tried to clarify for the relation between Darwinian evolution and Entropy production in the present manuscript.
The new last paragraph at the end of Section 9 may help to clarify the relation between the evolution of catalysts and entropy production. I have tried to clarify the relation between Darwinian evolution and entropy production in Section 10. In the first paragraph of Section 10, I have also added a reference to Woese 2002 [145] in case the idea of “the origin of Darwinism” is unfamiliar to the reader.
At the end of summary, in line 630-632, the author proposed "by a simple cream in coffee model an operational definition in which complexity is proportional to dS/dt".
Comments of the reviewer are mostly summarized by focusing on this sentence, although related descriptions are found all over the manuscript.
First of all, what part of entropy change the author means by dS/dt ? Far from equilibrium dissipative system (FFEDS) is generally understood as a local steady system in which a dynamical structure constantly creates entropy, and simultaneously keeps a low entropy state by constantly throwing out this produced entropy out into the reservoir. This is only possible when the local system is immersed in a low entropy reservoir (This is why it is called Far From Equilibrium.) which is open to accept the produced entropy for carrying out into the atmosphere. Low entropy local system with a stable dynamical structure which keeps producing entropy is the essential character of FFEDS, whether it is biotic or abiotic.
The Reviewer raises an important point. I agree with the reviewer’s description of what FFEDS are, with the possible exception that biogeochemical cycles ([88] Falkowski et al 2008) and Gaian regulatory mechanisms [64-67] may qualify as FFEDS but are not necessarily “local”.
I agree with the Reviewer that one should be as specific as possible about what dS/dt means. One effort is in the last two paragraphs of Section 2 where I describe several problems associated with attempts to find a unified definition of dS/dt while trying to “go beyond the second law”. One problem is the wide variety of contexts for which we would like to have a unified concept of entropy and entropy production. These contexts include (first paragraph of section 4):
“convection cells, hurricanes, whirlpools, lightning, stars and galactic bars. These are caused by gradients of various kinds; thermal, pressure, humidity, photochemical, electric, chemical redox and gravitational [54, 71].”
Unfortunately, we do not have a unified concept of entropy and entropy production in the wide variety of contexts for which we would like to have one. This problem is explored and described in the work of Bruers [40-42] cited in the last paragraph of Section 2.
“Bruers [40 - 42] discusses various versions of MEP depending on whether the entropy production is near equilibrium or far from equilibrium. His “far-from-equilibrium variational MaxEP” seems to be the most relevant to keeping track of biological entropy production.”
The last two sentences in Section 2 also acknowledge this issue:
“The validity and applicability of these various entropy production principles seem to depend on the problem and on the mechanism of entropy production. Thus, currently, there is no general-purpose principle for systems far from thermodynamic equilibrium.”
This lack of a unified concept of entropy production is a particularly irksome problem for cosmologists who observe a maximum entropy blackbody cosmic microwave background in the early universe, but have to reconcile this observation with the 2nd law under which the universe had to start at low entropy. This problem is discussed in [54] (Lineweaver & Egan 2008).
We do not have a unified general definition of entropy and entropy production that would be applicable to the broad range of contexts for which we want to invoke the second law (and go beyond it). Thus, the discussion of entropy production in this broad range of contexts is necessarily qualitative.
The "cream in coffee model" is a closed system and difficult to be used for explaining FFEDS.
I used the cream in coffee model because it has been used multiple times by one of the most authoritative researchers who has addressed the issue of the relationship between complexity and entropy in a cosmic context: Sean Carroll [132,133]. I also used the cream in coffee model because the lowest panel in Fig. 2 shows an important conceptual and qualitative difference with what has been proposed previously.
Secondly, "complexity" is not defined in the manuscript. How can an undefined quantity is connected to dS/dt by a mathematical word "proportional"? Therefore, this sentence in the summary does not seem to be meaningful.
Reviewer believes that the present research will become fruitful in future, when the author clarifies how he thermodynamically defines entropy, whether he defines it for each biological body or for some groups or total biomass.
Also, it would be necessary for the author to define complexity after clarifying whether he wishes to study complexity of a biological body or for the ensemble of biologies.
Just as we have no unified concept of entropy and entropy production, we have no consensus definition of complexity. This is discussed in [94] Lineweaver et al 2013 Complexity and the Arrow of Time as well as in the new reference [95]: Lineweaver, Davies & Ruse, “What is complexity? Is it increasing?” In Section 9: “The Relationship between Entropy Production and Complexity” (just after Fig. 2) I acknowledge that complexity is hard to define, and then I suggest an operational definition of complexity and a way to test it:
“Although complexity is notoriously hard to define [94], a potentially useful operational definition is that the time-dependence of complexity is proportional to the time-dependence of entropy production: complexity(t) ~ dS/dt (t).”
I now specify that for this proposed test, I am referring to Kolmogorov algorithmic complexity and I cite [134] (Kolmogorov 1965). Thus, in the text just before Fig. 3 I have changed:
“To test which model is more correct one could take a video of this process and, using a file-compression algorithm, quantify the complexity of the turbulent swirls [127] p. 232.”
to
“To test which model is more correct one could take a video of this process and, using a file-compression algorithm, quantify the Kolmogorov algorithmic complexity of the turbulent swirls [133] p. 232, [134].”

Reviewer 2 Report (New Reviewer)
Comments and Suggestions for Authors
The author is well known for his contributions to thermodynamics. However, thermodynamics is not the explanation for every chemical phenomenon, even though there is no argument that every physical/chemical phenomenon is consistent with the laws of thermodynamics. However, the author seems to overlook the fact that chemical phenomena are governed not just by thermodynamic laws but by kinetic ones as well. Alfred Lotka was already aware of that in his classic paper in PNAS published 100 years ago (doi: 10.1073/pnas.8.6.151). The author found it convenient to quote the adjacent thermodynamic paper (ref 19) but not the kinetic one which is the central one for evolutionary understanding. As Lotka says: "The two fundamental laws of thermodynamics are, of course, insufficient to determine the course of events in a physical system. They tell us that certain things cannot happen, but they do not tell us what does happen. " I think that Lotka's statement says it all.
In any case it is puzzling to me that a paper on Darwinian evolution does not mention the word replication once. I find it problematic that the author discusses the topic of evolution without mentioning the kinetic power of replication which is clearly central to Darwinian evolution. I would suggest that as a minimum that the author consider the possible role of kinetics to the observed phenomenon of increasing complexity in evolution (which I suspect is the true explanation) and offer reasons why he believes that thermodynamics appears to him to be the more appropriate explanation
Author Response
Please see attached file.

Reviewer 3 Report (New Reviewer)
Comments and Suggestions for Authors
This article investigates the role of entropy production, and its temporal evolution, as a unifying principle that could extend our understanding of evolution beyond the traditional Darwinian framework. The author poses a profound and stimulating challenge to classical Darwinian concepts such as evolutionary pressure and especially to the assumption of a continuous increase in biological complexity, pointing out that these are not universally observed. By grounding the analysis in the second law of thermodynamics and emphasising the generality of entropy and its production, the article proposes a broader theoretical basis that could overcome some of the limitations of Darwinian evolution, offering new insights into the origin and development of life.
I thoroughly enjoyed reading this article and believe it is of great value to a wide audience, from biologists to mathematicians. The author effectively positions entropy production as a central unifying quantity that connects diverse concepts and theories, emphasising that ultimately the dynamics of energy and entropy governs the evolution of the universe itself. I have only minor comments for the author:
1. At the end of the first section, the author refers to a metaphor. This leads me to wonder: could life be considered a metastable non-equilibrium state that evolves, driven by entropy production or perhaps by a free energy–like potential that also depends on dissipation? Can the author briefly comment about this?
2. Line 172: This statement is not entirely accurate, as patterns do not necessarily require a system to be "far from equilibrium" to emerge. Being out of equilibrium is a sufficient condition. The notion of "far from equilibrium" can be misleading, especially when interpreted through the classical lens of the second law, entropy, and free energy. Such interpretations often require the introduction of information entropy or pseudo-thermodynamic quantities, which may complicate rather than clarify the understanding of self-organization and the origin of life itself as well as the evolution. This “limitation” is not essential for understanding these phenomena.
3. In the last paragraph of Section 3, I found the argument particularly compelling. Has the author considered that both life and the environment interact to play their respective roles in entropy production, perhaps in the spirit of Gaia theory? If so, what is the relevant magnitude guiding this interaction? Is it the maximum entropy they can produce? The minimum required to maintain a minimum of lost work or free energy? Or could it be an intermediate state governed by a dissipative thermodynamic potential yet to be defined? What could the author comment on this?
4. In Section 7, I found the introduction of Gollo’s law somewhat difficult to follow. Could the author consider presenting this law in a way that allows non-expert readers to grasp the main idea without feeling that key context or explanation is missing?
5. Could the author propose a more quantitative formalism—perhaps a specific equation—to allow expert readers to explore the idea in greater depth? If not, could the author refer to previous works beyond those of Prigogine, Dewar, and Bejan that have proposed thermodynamic frameworks for the emergence of self-organized structures or even the origin of life?
6. Has the author considered including a brief section discussing the definition of life? I understand this is a challenging task, but when treating the evolution of life as a consequence of entropy production, such a definition might become more general, or at least differ from conventional ones. This could potentially avoid the need for the discussion around the "wall of minimum complexity." In other words, what we currently recognize as living beings might be just a particular case within a broader class of self-organized complex structures.
7. At the end of Section 9, I am once again reminded of the importance of going beyond simply considering entropy production or its increase. For instance, works such as Arango-Restrepo et al., Phys. Chem. Chem. Phys., 21(32), 17475–17493 (2019), and J. Phys. Chem. B, 125(7), 1838–1845 (2021), explore the idea that self-organized structures and patterns emerge at specific values of the entropy production rate, which depend on parameters characterizing their shape or functionality. This perspective might be valuable here, as ongoing efforts in the fields of complex systems and active matter are now actively seeking to quantify and relate entropy production to the emergence of self-organization.
Overall, this is exceptional work. I will gladly recommend its publication once the author addresses the comments.
Round 2
Reviewer 1 Report (New Reviewer)
Comments and Suggestions for Authors
Ambiguity of the concepts and terminologies which the author discuss was not clarified in the revised manuscript.
The author only cited some more references which explained the difficulties or ambiguities for clear definition of the concept. It is always difficult to define some concept in a way to explain complex phenomena globally. But scientific progress has been made by a creative research by building a new concept for explain some specific phenomena and by trying to examine if it is applicable to wider range of phenomena.
In section 9, the author suggested an operational definition of complexity and a way to test it using Figure 2 of cream in coffee model. The author should have shown the experimental results and compare it with Kolmogorov’s definition, not only suggesting the idea.
However, there is another problem for the author’s operational definition; complexity (t)~ dS/dt., the quantity S is the total entropy of a closed system, and therefore, dS/dt is not independent of S of the closed system. As the reviewer pointed out in the first Review, FFEDS(Far from equilibrium dissipative system) is an open system which allows two independent quantities; entropy S of a local system and entropy production σ which is not dS/dt. This difference is created by the internal dissipative structure and flow out of the produced entropy to outside of the system. Without this mechanism dissipative structure is not realized, either biotic nor a biotic.

Author Response
Please see attached file.

Reviewer 2 Report (New Reviewer)
Comments and Suggestions for Authors
Though I believe the thermodynamic thrust of this paper is not the correct one, I will allow it to be published enabling readers can make up their own minds on the issue.
Author Response
Please see attached file.

Reviewer 3 Report (New Reviewer)
Comments and Suggestions for Authors
The author has successfully addressed my comments.
Round 3
Reviewer 1 Report (New Reviewer)
Comments and Suggestions for Authors Third Comments by Reviewer,
There have been published papers which discussed on the possibility of
explaining the evolution of biological world in terms of entropy
production, as were referred in the present manuscript.
The author of the present paper tried to connect the entropy production
with complexity of the Darwinian evolution. Because Darwin had never
quantified his theoretical statements such as “survival of the fittest”,
it should be useful for the development of evolution theory if the
author could have shown the relation between the complexity of
bio-system and the entropy production.
Unfortunately, the author simply assumed that the entropy production
of his definition is equal to the complexity which is not defined in the
manuscript. Therefore, the paper does not contain a clear scientific
assertion in the present form.
There are two fundamental problems which the author has to overcome.
First is the definition of the complexity. Some opinion may insist,
contrary to the author’s proposal, that the entropy production is only a
quantity, while negative entropy itself is related to information of the
structure. It seems difficult to persuade these opinions by the present
manuscript.
Second is the system which the author discusses. In some section the
author discuss the entropy and its time derivative of the whole
universe, and in some section the author discuss FFEDS which is local.
Complexity which the author wishes to define is the complexity of the
universe, not the complexity of Darwinian world. It seems difficult to
for the readers of the Journal to understand this relation.
The reviewer judges the paper has not yet reached the standard for
publication.
Round 4
Reviewer 1 Report (New Reviewer)
Comments and Suggestions for Authors
The manuscript was improved by weakening the expression between the entropy production and complexity.
The whole text of the present manuscript is still descriptive and qualitative, and therefore does not appeal strongly to the reader.
Complexity is fundamentally a set of quantities deduced from the spatio-temporal pattern of the object, and is naturally depends on the spatio-temporal scale. Therefore, complexity of bio-system is not easy to obtain, compared to coffee with milk.
The reviewer wishes the author to make in future a quantitative progress in this subject.
This manuscript is a resubmission of an earlier submission. The following is a list of the peer review reports and author responses from that submission.
Round 1
Reviewer 1 Report
Comments and Suggestions for Authors
"Darwinian Evolution as a Product of the Second Law" is an ambitious review of life, evolution, the universe, and everything. It positions a tendency for entropy production to increase at the central position. Through the manuscript, the author offers more than ten citations of sentences and paragraphs from books and papers, including his publication from 2008. The main focus of the manuscript is the author's struggle with the hypothesis that goes beyond the second law. The initial definition of the hypothesis, named 2B, refers to the whole universe. The author proposes that entropy change dS/dt is entropy production and that entropy production of the universe tends to a maximum. Close to the end of the manuscript, the author admits that "cosmic entropy production does not conform to hypothesis 2B." The author's solution is adding a free-energy-availability constraint and renaming hypothesis 2B'. In the middle of the manuscript (page 7), the author invokes the evolution of ecosystems and catalytic enzymes to suggest the interpretation of hypothesis 2B as an "entropically-driven increase in entropy production." The author explains the hypothesis as "the unifying concept of evolution as a tendency to increase dS/dt." The author opinioned: "We need to do a better job of convincing our colleagues of the fundamental constructive role entropy plays in the origin of life and Darwinian evolution." However, the manuscript does the opposite. In its present form, it goes in too many directions, none with solid foundations nor convincing conclusions. The author readily admits that quantitative statements on how life forms enhance the operation of the second law are a work in progress. However, the author's contribution does not qualify as research progress. My recommendation is to reject the manuscript.
Specific comments:
The erroneous equalization of entropy change dS/dt in the system with the entropy production diS/dt due to irreversible processes should not be published in any scientific journal.
In all my experience as a reviewer, I have never seen a worse shape of references. Several references mentioned in the main text are not present in the list of references (Vallino 2014; Toniazzo et al 2005). A single reference in the list of references is often associated with two or three reference numbers (Castelle and Banfield 2018 with numbers 14 and 15; Cech 2012 with numbers 16 and 17; Hug et al 2016 with numbers 43, 44, and 45; Lineweaver 2020 with numbers 59 and 60; Lovelock and Margulis 1974 with numbers 61 and 62; Serjant et al 2010 with numbers 80 and 81; Weiss et al 2016 with numbers 85 and 86). There is no consistency in how references are cited. Sometimes, the publication year is mentioned after the authors' names, sometimes after the journal's name, and sometimes at the very end of the citation. Author names are often scrambled or changed (Martyshev instead of Martyushev is just one of many such examples).
Ideas about Kardashev Type civilizations: "Type III can access all the energy available from the hundreds of billions of stars in a galaxy (Kardashev 1964, 1997). With a much larger supply of free energy they/we will get bigger and figure out how to make anti-matter engines, build Dyson spheres and possibly whole universes" belong to magic or soft science fiction.
Reviewer 2 Report
Comments and Suggestions for Authors
This manuscript presents an interesting discussion on the link between entropy production and life evolution. In particular, it explains why several old ideas, that have emphasized the apparent contradiction between entropy growth and evolution, are plainly wrong. Therefore this is a useful contribution. I have however some comments that the author may consider before publication.
1/ My main disagreement with the views expressed in this paper is the idea that the rate of entropy increase should “always” increase (equation 2B), something which is, by the way, disproven already on Figure 4: dS/dt does in fact decrease three times on this sketch (after reheating, after SMBH formation, and at the end): the final suggestion that there is an ultimate entropy value Smax explains only the latest decrease, not the other ones.
More generally, the rate dS/dt decreases when the gradients are getting smaller or disappearing: a fire stops when there is nothing left to burn; stars do die in the same way. The same is true for life: death happens when there is nothing to eat and entropy then increases at a slower rate. The same is also true for evolution: the story is very likely not linear, but underwent many ups and downs… In the same fashion, paragraph 9 (techno FFEDS) is overly “optimistic”: another likely possibility is that civilizations may develop “too fast” and forget to maintain some vital gradients (which is not only a question of sources, but also a question of waste management). Overall, in Darwinian evolution, a key concept is “trial and error” (with many errors and few successes), something based on randomness and accumulation of information. Survival of the fittest is possible only when there are many (more than one!) specimens or trials, which is something less obvious when talking about planets, civilizations or the whole universe.
In other words, I may (to some extent, see below) agree with the view that “complexity” is linked in some way to the efficiency of a system to dissipate free energy (“complexity” = dS/dt), but certainly not with the view that it should necessarily always increase with time… It might be true at some time periods, but wrong at other periods. It might be true for some specific civilization on a remote planet, but wrong for others. It might be true in an “ensemble average” of all possibilities, but not necessarily in a specific (our) universe…
2/ The author insists that “life” and “non-life” should obey the same principles (for instance, in the 1st part, he says: “So, we will necessarily have to deconstruct the current apparent boundary between non-life and life to understand the transition from chemistry to biology). But at the end of the paper, he makes a rather strong distinction between « biotic » and « abiotic » FFEDS, or with a universe with or without life. This appears quite contradictory to me… Or, maybe the author could expand a bit the discussion on the necessity of “information storage” to this acceleration process: RNA/DNA for life evolution, brain/culture for mankind and social evolution, IA for possible speculations about the future.
3/ The author also insists on life being more “efficient” to deplete gradients. For instance (part 6) “the evolution of catalytic enzymes and the phylogenetic divergence of species, produce a variety of dS/dt values. The larger ones give more efficient access to free energy and/or access to a larger number of sources of free energy ». But for living organisms, information is not only about “efficiency” to use gradients, but mostly about sustainability, ie. about NOT using all the food at the fastest possible rate, but on the contrary doing so for the longest possible duration. For illustration, this is the main difference between fires and mushrooms when oxidizing wood: the former are burning fast and die rapidly without offspring, the latter live much longer and generously transmit their genes to future generations (that may possibly feed on the same wood). Clearly, ∆t being much smaller, dS/dt is larger for individual fires than for mushrooms, but overall indeed, fires are less common than mushrooms on our planet, which therefore better contribute to the overall entropy increase. Sustainability is certainly much more critical than efficiency.
Reviewer 3 Report
Comments and Suggestions for Authors
Interesting and well-focussed indeed, but too much info still required.
